# Validation of the Ambivalence and Uncertainty Scale

**DOI:** 10.3390/ijerph23010046

**Published:** 2025-12-29

**Authors:** Julia-Katharina Matthias, Andreas M. Baranowski, Anna C. Culmann, Simone C. Tüttenberg, Yesim Erim, Eva Morawa, Petra Beschoner, Lucia Jerg-Bretzke, Christian Albus, Sabine Mogwitz, Franziska Geiser

**Affiliations:** 1Klinik und Poliklinik für Psychosomatische Medizin und Psychotherapie, Universitätsklinikum Bonn, Rheinische Friedrich-Wilhelms-Universität Bonn, 53127 Bonn, Germany; andreas.baranowski@ukbonn.de (A.M.B.); anna.culmann@ukbonn.de (A.C.C.); simone.tuettenberg@ukbonn.de (S.C.T.); franziska.geiser@ukbonn.de (F.G.); 2Psychosomatische und Psychotherapeutische Abteilung, Universitätsklinikum Erlangen, Friedrich-Alexander-Universität Erlangen-Nürnberg (FAU), 91054 Erlangen, Germany; yesim.erim@uk-erlangen.de (Y.E.); eva.morawa@uk-erlangen.de (E.M.); 3Klinik für Psychosomatische Medizin und Psychotherapie, Klinikum Christophsbad Göppingen, 73035 Göppingen, Germany; petra.beschoner@uniklinik-ulm.de; 4Klinik für Psychosomatische Medizin und Psychotherapie, Universitätsklinikum Ulm, 89081 Ulm, Germany; lucia.bretzke@uni-ulm.de; 5Klinik und Poliklinik für Psychosomatik und Psychotherapie, Universitätsklinikum Köln, 50937 Köln, Germany; christian.albus@uk-koeln.de; 6Klinik und Poliklinik für Psychotherapie und Psychosomatik, Universitätsklinikum Carl Gustav Carus Dresden, 01307 Dresden, Germany; sabine.mogwitz@ukdd.de

**Keywords:** ambivalence, validation, scale, psychology, COVID-19

## Abstract

**Background**: This study aimed to validate the Ambivalence and Uncertainty Scale (AUS), developed to measure dispositional ambivalence, ambivalence intolerance, and decision-making difficulties, particularly among healthcare professionals during high-stress situations such as the COVID-19 pandemic. **Method**: Cross-sectional data from 1240 German healthcare professionals were analyzed. Exploratory factor analysis (EFA) was employed to evaluate the scale’s dimensionality, while internal consistency and construct validity, including convergent and divergent validity, were assessed using correlations with relevant psychological constructs. **Results**: The EFA revealed a unidimensional structure explaining 64.33% of variance, indicating a single underlying trait. The AUS demonstrated excellent internal consistency (Cronbach’s α = 0.86) and strong convergent validity, evidenced by significant positive correlations with anxiety and depressive symptoms (r = 0.63) and burnout (r = 0.48), and a negative correlation with sense of coherence (r = −0.60). Divergent validity was supported through minimal correlation with unrelated constructs such as fatigue (r = −0.02) and a moderate correlation with work–family conflict (r = 0.31). The AUS effectively captures dispositional ambivalence and uncertainty tolerance, highlighting its relevance in psychological adaptation and resilience among professionals in stressful environments.

## 1. Introduction

During the COVID-19 pandemic, healthcare professionals were frequently confronted with highly demanding and emotionally conflicting situations. These included balancing the delivery of optimal patient care with the reality of limited resources, managing the fear of infection while continuing to provide care for infected patients, and grappling with personal concerns about family safety alongside professional responsibilities [1,2,3,4]. Such scenarios are prime examples of experiences that can evoke ambivalence [5].

Ambivalence, as defined in psychology, refers to a state of emotional or cognitive conflict arising from the coexistence of opposing thoughts, feelings, or inclinations toward a specific object, idea, or decision [6]. This construct is relevant across various domains of human experience, including decision-making, emotional regulation, and social interactions. Ambivalence is not merely a transient state of indecision but an essential aspect of psychological functioning, with implications for both well-being and stress management. For instance, high levels of ambivalence can complicate decision-making processes, prolonging the time needed to reach conclusions and increasing feelings of uncertainty [7]. This persistent emotional conflict may exacerbate tension and overwhelm, potentially contributing to chronic psychological strain [8].

Despite its challenges, ambivalence is not inherently detrimental and can serve as a catalyst for adaptive coping. By encouraging reflective evaluation of complex experiences, ambivalence fosters an understanding of conflicting perspectives. This process may promote resilience and meaning-making, particularly in the face of crises, by helping individuals adapt to adversity and find balance amidst uncertainty [9]. In the discourse on resilience, ambivalence and the ability to tolerate conflicting emotions—termed ambivalence tolerance—has been identified as integral to navigating crises and adapting to adversity [10].

To date, research on ambivalence has primarily focused on characteristics of the underlying attitudes [10,11] or the contextual factors influencing evaluations, as a means of understanding the strength and effects of ambivalent feelings [12,13,14,15]. More recently, increasing attention has been directed toward individual differences in ambivalence. Schneider et al. [16] introduced the concept of trait ambivalence, which refers to a generalized tendency to experience conflicting emotions or thoughts across various situations. Understanding trait ambivalence enables researchers to account for interindividual differences in attitudinal ambivalence by identifying baseline tendencies, thereby disentangling stable personality traits from context-specific evaluative conflicts.

In the context of the COVID-19 pandemic, it became evident that trait ambivalence extends beyond the mere experience of conflicting thoughts and emotions. In high-stress environments, such as healthcare settings, individuals not only tend to experience inner contradictions but also differ in their ability to tolerate these conflicts and manage uncertainty [17]. The ability to endure and integrate these tensions represents an important facet of trait ambivalence, influencing how individuals navigate complex emotional and cognitive conflicts across situations. Moreover, decision-making difficulties [18,19] and intolerance of uncertainty [20] appear to be stable dispositional tendencies that shape how people respond to ambivalence in everyday life. Thus, to fully understand trait ambivalence as an individual difference, it is necessary to assess not only the frequency and intensity of ambivalent experiences but also how well individuals tolerate and regulate them as part of their personality structure.

To address this broader conceptualization, we developed the Ambivalence and Uncertainty Scale, a trait-based instrument designed to capture multiple facets of dispositional ambivalence. By integrating measures of ambivalence intolerance and decision-making difficulties alongside the fundamental experience of ambivalence, this scale provides a more comprehensive assessment of how individuals characteristically react to conflicting emotions and uncertainty. Such an approach is particularly relevant in contexts characterized by high levels of chronic uncertainty, such as global pandemics, emergency response settings, and other high-pressure environments where individuals’ stable tendencies toward ambivalence and uncertainty tolerance influence their ability to cope. Beyond healthcare, this instrument may contribute to a better understanding of individual differences in resilience and cognitive-emotional regulation among professionals regularly facing high-stakes decision-making, such as first responders, law enforcement officers, and military personnel. In addition to psychological and situational factors, neurobiological principles are increasingly being considered to better understand how individuals experience and cope with ambivalence and uncertainty. This approach focuses on neurobiological systems, such as the Behavioral Inhibition System (BIS), that specialize in processing conflict and uncertainty. Within this framework, ambivalence is conceptualized as a value-neutral, trait-like disposition; the scale assesses this dispositional core, while context-dependent outcomes—costs (stress, decisional delay) and benefits (reflection, perspective integration)—are treated as correlates outside the construct.

The aim of this study was therefore to develop and validate the Ambivalence and Uncertainty Scale and examine its psychometric properties in a sample of healthcare professionals. To guide validation, we pre-specified a nomological network and hypotheses for convergent and divergent relations with established constructs; full details of the strategy and measures appear in the Methods.

## 2. Materials and Methods

### 2.1. Data Collection

This study was part of a larger multicenter longitudinal research project on stress and resilience among employees in clinical healthcare or enrolled in accredited clinical training for a healthcare profession (e.g., medical students during rotation, nursing trainees) [21]. The present analysis utilized cross-sectional data from a single time point within this longitudinal framework. Data were collected through an online survey conducted between April and July 2023. Because the parent multicenter study scheduled assessment waves approximately every 9–12 months, no short-interval re-test window (e.g., several weeks) was feasible within the protocol; accordingly, test–retest reliability could not be estimated in the present dataset.

Participants were recruited via internal mailing lists of the University Hospital Bonn, several general hospitals, and professional medical associations. Inclusion criteria required participants to be at least 18 years old, employed in clinical healthcare, residing or working in Germany, and proficient in the German language. The survey was conducted using the online platform Unipark (www.unipark.com) and took approximately 20–30 min to complete. Ethics approval for the study was obtained from the Ethics Committee of the Medical Faculty of the University of Bonn (Reference Number: 125/20). All participants provided informed consent prior to participation.

### 2.2. Development of the Ambivalence and Uncertainty Scale (AUS)

Given our understanding of ambivalence as a trait-like disposition that influences emotional regulation, decision-making, and uncertainty tolerance, we sought to develop a more comprehensive instrument to assess trait ambivalence. While existing measures capture aspects of evaluative conflict [6,10] and attitudinal ambivalence [16], they do not sufficiently account for the integration of ambivalence experience, ambivalence intolerance, and decision-making difficulties. To bridge this gap, we aimed to design a scale that better reflects the dispositional capacity to experience, tolerate, and regulate conflicting emotions and uncertainty, particularly in high-stress environments such as healthcare.

Our research group, comprised experienced psychologists, physicians, and theologians, developed the Ambivalence and Uncertainty Scale using a systematic approach. We combined existing validated measures and adapted them as necessary, while also developing new items to ensure a comprehensive assessment of ambivalence-related constructs. Our goal was to create a scale that captures three core facets:Experience of Ambivalence—the tendency to frequently experience contradictory emotions and thoughts.Ambivalence Intolerance—the discomfort associated with holding conflicting attitudes and the tendency to avoid or resolve ambivalence.Decision-Making Difficulties—the dispositional tendency to struggle with choices, particularly under conditions of uncertainty.

Importantly, we operationalized ambivalence intolerance as a trait-like regulation tendency rather than symptomatic strain. Items were deliberately framed without state-stress terminology and without references to context-bound outcomes (e.g., “stress,” “burnout”), targeting a preference for rapid resolution/avoidance of evaluative conflict. Thus, intolerance indexes how individuals process ambivalence, whereas downstream reactions are treated as external correlates in the nomological network.

Following a literature review of established ambivalence and uncertainty measures, we derived three core items from existing validated scales that reflected each of these dimensions: “I experience my feelings as contradictory” (Experience of Ambivalence), “I find it difficult to tolerate contradictory emotions or thoughts” (Ambivalence Intolerance), and “I find it easy to deal with uncertainty and ambiguity” (Decision-Making Difficulties (reverse-coded)).

These themes align with previous measures of trait ambivalence, including the Trait Ambivalence Scale [16] and subjective ambivalence items introduced by Prister [11], which assess perceived evaluative conflict and affective discomfort. Furthermore, they fit with constructs from the Intolerance of Uncertainty Scale [21].

To further expand the scale and include a biologically grounded approach to ambivalence and uncertainty, we adapted validated items from the Behavioral Inhibition System (BIS) scale of the Reinforcement Sensitivity Theory of Personality Questionnaire [22]. The BIS is a neurobiological system responsible for processing uncertainty, conflict, and risk aversion, making it an ideal complement to the AUS in assessing individual differences in ambivalence tolerance and decision-making difficulties. Four items were chosen to reflect decision-making difficulties, two to represent the experience of ambivalence, and one to capture ambivalence intolerance, based on their conceptual fit with these constructs (for a list of all items, see Table 1 and Appendix A for the German original version). 

For each item, participants were asked to indicate how much each statement applied to them on a 4-point scale (1 = does not apply to me at all, 2 = rather does not apply, 3 = rather applies, 4 = fully applies). We opted for a 4-point forced-choice Likert format to minimize midpoint/central-tendency bias under field conditions, reduce respondent burden in a multi-instrument survey, and still ensure adequate reliability and discrimination while enabling appropriate ordinal estimation (polychoric Exploratory Factor Analysis [EFA], Confirmatory Factor Analysis [CFA], weighted least squares mean and variance adjusted estimator [WLSMV]).

The inclusion of BIS-related items ensures that the scale not only captures subjective and cognitive dimensions of ambivalence but also aligns with individual differences in neurobiological sensitivity to uncertainty. This provides a more robust and integrative framework for understanding trait ambivalence as a personality disposition.

Initially, we developed the Ambivalence and Uncertainty Scale (AUS) to integrate a broader conceptualization of trait ambivalence into our multicenter research on resilience among healthcare professionals. The scale was designed to assess the experience of ambivalence, ambivalence intolerance, and decision-making difficulties within the high-stress context of clinical work during the COVID-19 pandemic.

To enable its use beyond our initial research framework, we conducted the present validation study, using new data collected from our multicenter project to systematically examine the psychometric properties of the AUS. This ensures that the scale meets scientific standards for reliability and validity, making it a valuable instrument for future research on trait ambivalence, uncertainty tolerance, and decision-making in high-stress professions. Due to stringent survey-length constraints in the parent multicenter study, no existing ambivalence scales were administered; instead, we prioritized measures sampling the nomological network to evaluate convergent and divergent validity.

### 2.3. Validation Strategy

We evaluated construct validity by testing theory-driven associations between the Ambivalence and Uncertainty Scale (AUS) and conceptually related (convergent) and distinct (divergent) measures. We predicted positive links with indicators of negative affect and burnout, a negative link with sense of coherence (as a global orientation toward comprehensibility/manageability/meaning), small links with role-conflict stressors (work–family/family–work conflict), and near-zero association with fatigue, which reflects energy depletion rather than evaluative conflict.

### 2.4. Additional Measures

#### 2.4.1. Depressive and Anxiety Symptoms (PHQ-4)

Trait ambivalence involves frequent co-occurrence of contradictory affect and uncertainty, which is theoretically tied to elevated negative affectivity (worry, dysphoria). The PHQ-4 indexes core anxiety/depressive symptoms that should covary with difficulty resolving conflictual appraisals, yielding a positive association. Convergence here supports the affective component of the AUS without implying construct redundancy.

Symptoms of depression and generalized anxiety during the prior two weeks were assessed using the Patient Health Questionnaire PHQ-4 [23]. This four-item questionnaire includes two items each for depression and anxiety symptoms, such as “Over the last two weeks, how often have you been bothered by feeling nervous, anxious or on edge?” and “Over the last two weeks, how often have you been bothered by feeling down, depressed, or hopeless?”. Responses were recorded on a Likert scale ranging from 0 (“not at all”) to 3 (“almost every day”). Higher scores indicate greater symptom severity. The total scores ranged from 0 to 12, with a cut-off value of ≥6 indicating cases of clinical depression and anxiety. Cronbach’s α in this sample was 0.83.

#### 2.4.2. Maslach Burnout Inventory—Human Services Survey for Medical Personnel (MBI-HSS (MP))

Burnout—especially emotional exhaustion—captures chronic strain that can arise when unresolved evaluative conflict and uncertainty accumulate in demanding roles. We therefore expected a positive, moderate correlation: related through shared strain and decisional load, yet distinct because burnout is contextual and occupational, whereas the AUS targets a trait-like disposition across situations.

Six items of the Maslach Burnout Inventory (MBI-HSS (MP)) were selected and adapted for this study to index burnout symptoms from the emotional exhaustion and depersonalization subscales; the personal accomplishment dimension was not included in this short form [24]. Example item includes “I feel emotionally drained from my work.” Responses were recorded on a six-point Likert scale from 1 (“never”) to 6 (“very often”), with higher scores indicating greater burnout. In this sample, Cronbach’s α was 0.82.

#### 2.4.3. Sense of Coherence (SOC-3)

Sense of coherence reflects viewing life as comprehensible, manageable, and meaningful; this orientation should buffer the experience of unresolved conflict and uncertainty. Accordingly, we predicted a negative association with the AUS, supporting discriminant polarity within a coherent nomological space (high SOC means lower dispositional ambivalence).

Sense of coherence was assessed using a German ultra-short version SOC-3 [25] of the original SOC scale [26]. This scale measures the extent to which individuals perceive life as comprehensible, manageable, and meaningful. The SOC-3 includes three items, such as “Do you feel you are in an unfamiliar situation and don’t know what to do?” Responses were recorded on a seven-point Likert scale, with higher scores reflecting a stronger sense of coherence (with the opening instruction “To what extent do the following statements and questions apply to you in the past two weeks?”). In this sample, Cronbach’s α was 0.70.

#### 2.4.4. Multidimensional Fatigue Inventory (MFI-20)

Subjective fatigue indexes energy depletion and reduced activation, not evaluative conflict or uncertainty processing. Aside from incidental overlap via general distress, theory predicts near-zero association with the AUS, supporting divergent validity.

Fatigue was assessed using the German version of the Multidimensional Fatigue Inventory [26,27,28], a 20-item self-report questionnaire measuring five dimensions of fatigue: general fatigue, physical fatigue, mental fatigue, reduced motivation, and reduced activity. In this study, four items were included to represent key aspects of fatigue across two dimensions: general fatigue and physical fatigue. Participants responded on a five-point Likert scale ranging from 1 (“yes, that is true”) to 5 (“no, that is not true”). An example item is “I feel tired.” All items were reverse-coded where necessary so that higher scores indicated greater fatigue. Cronbach’s α for the total scale in this sample was 0.74.

#### 2.4.5. Work-Family Conflict (WFC)/Family-Work Conflict (FWC)

WFC/FWC are domain-specific role-interference stressors driven by time, strain, and behavior-based incompatibilities between roles. While decision-related strain can contribute to interference, ambivalence, as measured by the AUS, is a general disposition to experience and regulate conflict across contexts. We therefore predicted small positive correlations: conceptually adjacent via strain, yet distinct from the broader, context-general ambivalence disposition.

Work-family conflict was assessed using a subset of four items from the German version of the Work-Family Conflict Scale [29]. The scale consists of two subscales: Work Family Conflict Scale (WFC) and Family Work Conflict Scale (FWC). Each subscale captures the directional nature of the interference between work and family roles. Responses were recorded on a five-point Likert scale ranging from 1 (“strongly disagree”) to 5 (“strongly agree”). An example item for the WFC subscale is “The demands of my work interfere with my home and family life.”. An example item for the FWC subscale is “My family’s needs or my partner’s demands are in conflict with my professional activities.”.

Higher scores indicate greater conflict between work and family roles. This sample of the WFC scale demonstrated excellent reliability, as indicated by a Spearman–Brown coefficient of 0.89. The Spearman–Brown coefficient of the FWC Scale was 0.64, indicating moderate reliability.

#### 2.4.6. Statistical Analysis

All statistical analyses were conducted using IBM SPSS Statistics (Version 26) and R (Version 4.4.1). A significance level of *p* < 0.05 was applied to all tests. Exploratory Factor Analysis was conducted to identify the underlying factor structure of the Ambivalence and Uncertainty Scale, and confirmatory factor analysis was conducted to evaluate global fit of the retained structure. The suitability of the dataset was evaluated using the Kaiser-Meyer-Olkin (KMO) test and Bartlett’s Test of Sphericity. The adequacy of the factor solution was evaluated using several fit indices, including chi-square (χ^2^), Goodness-of-Fit Index (GFI), Adjusted Goodness-of-Fit Index (AGFI), Comparative Fit Index (CFI), Tucker–Lewis Index (TLI), Root Mean Square Error of Approximation (RMSEA), and Standardized Root Mean Square Residual (SRMR). A validity analysis was conducted using both convergent and divergent validity, which were assessed using Pearson’s correlation.

## 3. Results

### 3.1. Sample Characteristics

The total sample consisted of 1253 participants aged between 18 and over 70 years, with the largest proportion falling in the 51–60 age group (27.3%, *n* = 342). Participants were primarily women (74.8%, *n* = 928), followed by men (25.9%, *n* = 321), and four individuals who identified as diverse (0.3%). Participants worked in a wide range of healthcare-related professions, including physicians, nurses, psychologists, medical-technical assistants, and pastoral care workers.

**Table 1 ijerph-23-00046-t001:** Sample characteristics and Mean AUS scores across gender, age and professional groups.

Variable	Category	*n* (%)	*M*	*SD*
Gender	Male	321 (25.6)	2.19	0.55
	Female	928 (74.1)	2.24	58
	Divers	4 (0.3)	2.40	0.82
Age	18–30 years	209 (16.7)	2.40	0.58
	31–40 years	319 (25.5)	2.28	0.58
	41–50 years	271 (21.6)	2.24	0.60
	51–60 years	342 (27.3)	2.01	0.53
	>60 years	112 (8.9)	2.20	0.58
Professional group	Nursing staff	419 (33.4)	2.23	0.56
	Physicians	201 (16.0)	2.10	0.53
	Medical-technical staff	146 (11.7)	2.32	0.56
	Psychologists	39 (3.1)	2.17	0.56
	Therapeutical professions	35 (2.8)	2.15	0.55
	Chaplains	49 (3.9)	2.01	0.49
	Students	12 (1.0)	2.60	0.69
	Other professional groups	352 (28.1)	2.21	0.59

Notes. Empirical range for all items 1–4. Physicians = Physicians and medical psychotherapists. Medical-technical staff = Medical-technical staff/medical specialists. Psychologists = Psychologists, psychological psychotherapists, child and adolescent psychotherapists. Therapeutical professions = Physiotherapists, occupational therapists, music therapists, logo therapists. Chaplains = Hospital chaplains. Students = Students and trainees of any specialization. Other = Administrative staff with and without direct patient contact. Emergency medical services. Midwife. IT employee. Research assistant. Study nurse. Social services. Educator. Special needs teacher. Special education teacher.

Quality screening in the EFA (*n* = 1253) yielded the following per-rule rates: LongString = 70 (5.6%), IRV = 46 (3.7%), monotone pattern = 1 (<0.1%), negative person–total correlation = 138 (20.4%), and Mahalanobis outlier = 7 (1.0%). Overall, 199/1253 (11.0%) were flagged by ≥1 indicator, while 62/1253 (4.9%) met the ≥2 indicators criterion. Sensitivity analyses excluding ≥2-flag cases in the analysis did not alter conclusions.

An EFA was initially conducted to identify potential factor structures within our scale. The Kaiser-Meyer-Olkin (KMO) test was performed to assess the suitability of our sample for factor analysis. The KMO value was 0.89, indicating an excellent level of shared variance among items for factor analysis. Values above 0.8 are deemed good, and those above 0.9 are considered excellent. Given the 4-point ordinal response format, we estimated a polychoric correlation matrix and conducted the EFA using principal axis factoring (PAF).

Bartlett’s Test of Sphericity indicated that the correlation matrix was significantly different from an identity matrix (χ^2^ = 3827.03, *p* < 0.001), confirming that correlations among items were sufficient to proceed with factor analysis. The KMO and Bartlett’s results suggested that the dataset was well-suited for a robust factor analysis.

In the next step, anti-image correlations were examined, which measure the amount of variance in an item that is not shared with other items. Items with anti-image correlations below 0.30 are generally recommended for exclusion [30]. In our dataset, the reversed item “I find it easy to deal with uncertainty and ambiguity” showed correlations below 0.30 with all other items, leading to its exclusion from further analyses.

We conducted a further EFA to extract the main independent factors after excluding this item due to insufficient shared variance with the rest of the scale. The KMO measure was 0.86, and Bartlett’s test remained highly significant (χ^2^ = 2847.30, *p* < 0.001), indicating that the data were appropriate for EFA. Beyond KMO/Bartlett, we evaluated factor retention on the polychoric matrix using Parallel Analysis, Velicer’s MAP, and VSS; MAP (minimum = 1 factor, MAP_1_ = 0.038) converged on a unidimensional solution (details in Appendix A), so no rotation was applied. Only factors with eigenvalues ≥1 were considered for extraction [31,32]. The analysis resulted in a unifactorial structure explaining 64.33% of the total variance, supporting the appropriateness of a one-factor solution for this scale (Table 2). Cronbach’s α of the new scale was 0.86. A single factor with an eigenvalue ≥1 was retained; no rotation was applied because only one factor was extracted.

For transparency, we also explored 2- and 3-factor solutions (polychoric PAF; oblimin rotation) and provided the rotated loadings in the Appendix A; these models showed diffuse loading patterns with multiple cross-loadings and high factor intercorrelations, offering no benefit over the unidimensional solution (see Appendix A).

### 3.2. Confirmatory Factor Analysis (CFA)

To evaluate the global fit of the retained structure, we estimated a one-factor confirmatory factor analysis. In our analysis, the GFI was 0.98, indicating excellent model fit, with AGFI at 0.95, demonstrating a very good fit while accounting for model complexity. The CFI was 0.96, and the Tucker–Lewis Index (TLI) was 0.94, both suggesting very good model fit with values close to 1. The RMSEA value was 0.07, supporting an acceptable fit (values below 0.08 are generally considered adequate), and SRMR was 0.04, indicating a low average discrepancy between observed and predicted correlations. Overall, the results of the EFA indicate that the proposed factor structure is a good representation of the observed data, with strong support from multiple fit indices. As a robustness check, Appendix A reports a split-sample EFA → CFA cross-validation (polychoric PAF EFA in a 60% subsample; one-factor WLSMV CFA in the 40% holdout) with bootstrapped loadings and fit-index quantiles, which corroborates the unidimensional model.

### 3.3. Measurement Invariance Across Gender and Age

We examined multi-group CFA with ordered indicators (WLSMV, θ-parameterization; latent variance fixed to 1) to test configural, threshold, and metric (thresholds + loadings) invariance. Invariance decisions followed commonly recommended change-in-fit criteria (ΔCFI ≤ 0.010; ΔRMSEA ≤ 0.015) between successive models [36,37,38,39]. Given our ordinal indicators and WLSMV estimation, we implemented the recommended testing sequence and identification for ordered-categorical outcomes [38]. Testing measurement invariance ensures that AUS scores are comparable across demographic groups; without equivalent thresholds and loadings, any apparent gender- or age-related differences could be artefactual rather than substantive. For gender, we contrasted male vs. female and excluded “diverse” due to very small *n*. For age, we used the five recorded age groups; to avoid empty response categories with ordinal WLSMV, adjacent categories were collapsed group-consistently where necessary (no cells with zero frequency remained). Invariance decisions followed common change-fit criteria (ΔCFI ≤ 0.010; ΔRMSEA ≤ 0.015 between successive steps).

For gender, fit remained stable from configural to thresholds and metric: ΔCFI ≤ 0.001; ΔRMSEA ≤ 0.015. For age, fit changes likewise stayed within recommended bounds (ΔCFI ≤ 0.002; ΔRMSEA ≤ 0.013), supporting equal thresholds and loadings across age groups (Appendix A for detailed statistics). Although absolute RMSEA values were modestly elevated—common for single-factor ordinal models with low df—the invariance conclusions rest on Δ-criteria, which clearly supported invariance. The statistical details can be found in Appendix A.

### 3.4. Convergent Validity

Convergent validity assesses the extent to which our scale correlates with other measures that are theoretically related. We hypothesized that ambivalence would correlate positively with constructs related to negative affect, such as anxiety and depressive symptoms. The PHQ-4, which measures symptoms of anxiety and depression, was used to test this hypothesis, given the conceptual overlap in negative emotional experiences. The results indicated a strong positive correlation between ambivalence and PHQ-4 scores (r = 0.63, *p* < 0.001), suggesting that both scales measure related constructs. Furthermore, we assessed the relationship between ambivalence and the MBI, which captures burnout symptoms, including depressive aspects. A moderate positive correlation was found (r = 0.48, *p* < 0.001), supporting the convergent validity of the scale by demonstrating associations with related indicators of psychological distress. In contrast, the SOC scale, which reflects an individual’s perception of life as comprehensible, manageable, and meaningful, was hypothesized to be negatively correlated with ambivalence. The results showed a strong negative correlation (r = −0.60, *p* < 0.001), indicating that individuals with a greater sense of coherence are less likely to experience ambivalence.

### 3.5. Divergent Validity

Divergent validity was assessed to determine whether our scale showed low correlations with measures of unrelated constructs, thereby supporting the distinctiveness of the construct. Divergent validity of the AUS was supported by low and non-significant correlations with the theoretically distinct construct of subjective fatigue, as measured by the MFI-20 (r = −0.02, *p* = 0.526). Furthermore, a modest correlation was observed with work-family conflict, WFC (r = 0.31, *p* < 0.001) and FWC (r = 0.28, *p* < 0.001), both domain-specific stressors that share some overlap with decision-related strain but conceptually differ from the broader psychological experience of ambivalence. These findings support the assumption that the AUS captures a distinct psychological construct, rather than general strain or fatigue (for a correlation table, see Figure 1).

## 4. Discussion

The present study aimed to validate the Ambivalence and Uncertainty Scale (AUS) by examining its psychometric properties and its relationship with related psychological constructs. Our findings confirm that the AUS demonstrates satisfactory internal consistency and construct validity. This suggests that the scale is a reliable and valid instrument for assessing dispositional ambivalence and uncertainty tolerance, at least within the population of German healthcare professionals facing the challenges of a pandemic. The results revealed a unifactorial structure, demonstrating that the items measuring uncertainty and ambivalence are closely interlinked and might reflect a more general underlying trait. Contrary to our initial conceptualization of three distinct facets—experience of ambivalence, ambivalence intolerance, and decision-making difficulties—the factor analysis indicated that these components do not emerge as clearly separable dimensions, but rather load onto a single latent construct.

Factor analyses demonstrated excellent fit indices and good factor loadings, confirming the internal consistency and structural validity of the scale. Furthermore, correlations with established psychological distress measures such as burnout symptoms, anxiety, and depression, along with negative correlations with resilience-related constructs like hope and sense of coherence, confirm the assumption that ambivalence experience and low ambivalence tolerance are related to negative affectivity and are relevant in processes of emotional regulation and psychological strain, particularly in stressful and uncertain contexts [36,37].

A major strength of the current study lies in its large and diverse sample size of healthcare professionals, offering relevance across various clinical roles. Additionally, the scale’s compact format facilitates practical implementation in both research and clinical settings, allowing efficient assessment of ambivalence as a risk factor in psychological health. Compared with prior tools that emphasize evaluative conflict alone, the AUS provides a compact dispositional profile spanning experience, tolerance/coping, and decisional regulation, supported by solid psychometrics in this sample. A major strength of the current study lies in its large and diverse sample of healthcare professionals, offering relevance across clinical roles. Additionally, the scale’s compact format facilitates practical implementation in research and clinical settings. Despite these strengths, several limitations should be noted. First, the exclusive reliance on a German-speaking healthcare sample during the COVID-19 pandemic may limit generalizability to other occupational groups and cultural contexts. Second, the parent study’s wave spacing (~9–12 months) precluded a short-interval re-test, so test–retest reliability could not be estimated. Third, although the AUS targets a trait-like disposition, cross-sectional self-report may blend dispositional tendencies with momentary affect and decisional strain, yielding partial trait–state overlap. Fourth, due to survey-length constraints, we did not administer existing ambivalence scales; consequently, direct head-to-head comparisons with prior instruments were not possible, which limits inferences about incremental validity relative to existing measures. Finally, we did not embed planned attention checks or timing screens; although post hoc indicators flagged a subset of responses (see Results), excluding ≥2-flag cases in the analysis did not alter conclusions, so that we retained the full sample to avoid unnecessary ad hoc exclusions.

Our sample consisted of healthcare professionals during the COVID-19 pandemic—a context marked by elevated uncertainty, role conflict, and ethical strain—so mean AUS scores and correlations with distress may be higher than in community samples. At the same time, the AUS items are domain-general (non-occupational) and the unifactorial structure with strong loadings suggests the construct is not setting-bound. We therefore expect structural validity to transfer, but population norms and measurement invariance should be established in non-clinical/community samples before making substantive between-group comparisons or setting cut-points.

The study’s exclusive reliance on a German-speaking healthcare professional population during the COVID-19 pandemic context could restrict its generalizability to other occupational groups and cultural backgrounds. Furthermore, the absence of test–retest reliability assessment poses limitations regarding conclusions about the temporal stability of trait ambivalence.

Future studies should address these limitations and extend the evidence base in several targeted ways: (1) establish temporal stability with short-interval test–retest designs (≈2–6 weeks) using ordinal-appropriate estimators; (2) examine incremental validity via head-to-head comparisons with existing ambivalence measures (e.g., hierarchical models) to quantify unique variance beyond evaluative conflict/attitudinal ambivalence; (3) evaluate cross-context generalizability through translations and multi-group measurement invariance across occupations, cultures, and languages (including DIF/MIMIC checks); (4) test predictive validity in prospective cohorts (e.g., forecasting stress responses, burnout trajectories, and decision performance under uncertainty); (5) assess intervention sensitivity, asking whether AUS scores change following decision-making or tolerance-of-ambiguity trainings; (6) develop IRT-informed short forms and reference values (norms, provisional cut-points) for research and clinical screening; and (7) triangulate AUS with behavioral and physiological indices from conflict/ambiguity paradigms to anchor the construct across methods. These steps will clarify the scope, utility, and boundaries of the AUS and help translate it into applied settings.

This study provides evidence for the validity and internal consistency of the AUS as a tool for assessing dispositional ambivalence in high-stress environments. Our findings confirm expected associations between trait ambivalence, psychological distress, and resilience-related constructs, which are relevant to the understanding of individual differences in cognitive-emotional adaptation. By offering a validated measure of ambivalence and uncertainty tolerance, the AUS provides researchers and clinicians with a valuable instrument for future investigations into decision-making, stress regulation, and psychological well-being.

## 5. Conclusions

This study validates the Ambivalence and Uncertainty Scale (AUS), developed to measure dispositional ambivalence, ambivalence intolerance, and decision-making difficulties, particularly among healthcare professionals facing high-stress environments, such as the COVID-19 pandemic. A cross-sectional sample of 1253 German healthcare workers participated, and exploratory factor analysis identified a single-factor solution accounting for 64.33% of the variance. The AUS demonstrated excellent internal consistency (Cronbach’s α = 0.86) and strong convergent validity, indicated by significant positive correlations with anxiety and depressive symptoms (r = 0.63), burnout (r = 0.48), and negative correlations with sense of coherence (r = −0.60). Divergent validity was supported by minimal correlations with unrelated constructs such as fatigue (r = −0.02). Results confirm that the AUS effectively captures dispositional ambivalence and tolerance for uncertainty, highlighting its relevance for understanding individual differences in emotional regulation and decision-making under stress. Future research should investigate the scale’s applicability across different occupational and cultural contexts and assess the temporal stability of the construct to further confirm its generalizability and practical utility. The AUS is therefore a reliable tool for studying ambivalence as a personality trait and its implications for psychological resilience and health.

## Figures and Tables

**Figure 1 ijerph-23-00046-f001:**
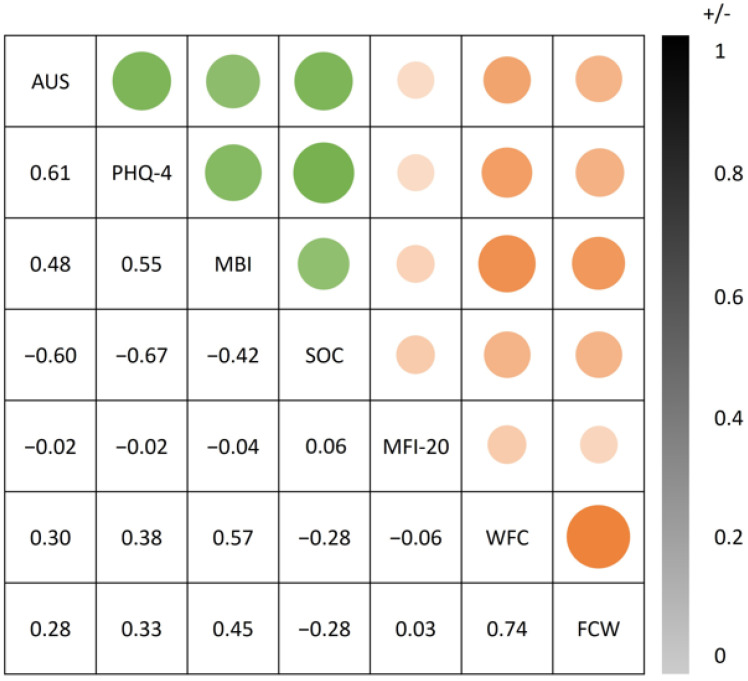
Bubble correlation plot of the Ambivalence and Uncertainty Scale (AUS). Correlations representing convergent validity are shown in green (PHQ = Patient Health Questionnaire; MBI = Maslach Burnout Inventory; SOC = Sense of Coherence), while correlations indicating divergent validity are shown in red (MFI = Multidimensional Fatigue Inventory; WFC = Work-Family Conflict). Bubble size and transparency reflect the magnitude of the correlation. The sign of the correlation is not visually distinguished but is indicated by the numerical values displayed next to the bubbles.

**Table 2 ijerph-23-00046-t002:** One factor solution for the Ambivalence and Uncertainty Scale.

Item	α If Item Deleted	ITC	Factor Loading	*M*	*SD*
1. I experience my feelings as contradictory	0.85	0.58	0.68	2.13	0.82
2. I find it difficult to tolerate contradictory emotions or thoughts	0.85	0.59	0.69	2.09	0.85
3. I am glad when decisions are made for me	0.85	0.52	0.62	2.24	0.83
4. I often doubt whether an effort is worth it	0.85	0.53	0.63	2.16	0.85
5. When choosing between attractive options, I find it difficult to decide	0.85	0.57	0.68	2.32	0.83
6. I find it difficult to tolerate uncertainty	0.85	0.57	0.67	2.56	0.86
7. I often do not know what I want	0.84	0.66	0.76	2.04	0.86
8. When I have to choose between two unpleasant alternatives, I find it difficult to decide	0.85	0.57	0.67	2.08	0.76
9. I often feel torn between different options or perspectives	0.83	72	0.81	2.20	0.84

Note. α if item deleted indicates the Cronbach’s alpha value if the respective item were removed from the scale. Cronbach’s α values between 0.70 and 0.95 are considered indicators of good internal consistency [33]. ITC = Item-Total Correlation, i.e., the correlation between each item and the total scale score (excluding the item itself), reflecting internal consistency. Values above 0.30 are generally considered acceptable [34]. Factor loadings represent the correlation between each item and the extracted factor. Loadings above 0.40 are considered acceptable, above 0.50 good, and above 0.70 very good [35].

## Data Availability

The data supporting the findings of this study are available upon reasonable request from the corresponding author, subject to compliance with institutional data-sharing policies and applicable privacy or ethical restrictions.

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
