# Peer review of "Validation of the Ambivalence and Uncertainty Scale"

_ijerph, 2025, doi:10.3390/ijerph23010046_

Round 1

Reviewer 1 Report

Comments and Suggestions for Authors
  1. In the Introduction, the construct is discussed ambiguously. For example, on p. 2 from line 55 (“For instance…”), ambivalence seems to be a psychological difficulty. Below, however (see p. 2 from line 65 [“Despite its challenges…”]), the construct seems to have positive meaning/characteristics. Similarly, on p. 2, from line 80 (The ability to endure…), ambivalence may be advantageous. Therefore, the reader at the end of the Intro is confused about your ideas about the construct, which, in the end, based on the emerging empirical evidence from your study, seems to have negative implications.
  2. In the Intro, you should also discuss the constructs with which you relate your new measure of ambivalence. For example, the reasons why work-family conflict and family-to-work conflict have been chosen are not clear. In a validation process, it is important to give reasons from the beginning about which factors, in the nomological network of a construct, will be examined and why. You should add a paragraph at the end of the Intro explaining in more detail your strategy.
  3. Data collection section. Why only cross-sectional data? It would be helpful to understand the stability of the measure, given that you conceptualize it as a stable factor. If you have longitudinal data, you should provide test-retest evidence. This is a critical piece of information regarding the validity of your new measure.
  4. Sample characteristics. Please include more information regarding sample composition related to professions. You just mentioned that the sample was made by nurses, psychologists, etc. You should provide more details.
  5. 4. By including in your measure aspects of ambivalence intolerance (i.e., discomfort), you risk putting together stressor (ambivalence feelings) and strain (reactions to those feelings), therefore contaminating the core of the ambivalence concept with its consequences. Can you clarify better why you adopted this choice? Additionally, including discomfort aspects suggests that you have a clear idea about the valence of the ambivalence construct – i.e., it is a negative phenomenon. If it is so, then your ideas about the implications of ambivalence should emerge better in the Introduction of your paper.
  6. It seems there are already ambivalence measures in the field. Why didn’t you collect data by using another ambivalence measure? This could have given chances to test your new measure against old measures, providing more persuasive evidence that your new measure captures something new and original.
  7. You used six items from the Maslach Burnout Inventory. Which specific sub-dimensions of burnout do they measure? You should provide more details.
  8. The item of SOC which you provide as an example seems a bit strange: “Do you feel you are in an unfamiliar situation and don’t know what to do?" Maybe you should check the translation? To make its meaning clearer, it could perhaps be helpful to provide the opening statement of the scale.  

  1. Goodness-of-fit statistics are usually provided with confirmatory factor analysis, which you did not carry out. You only focused on exploratory factor analysis. Relatedly, it is not clear whether you had to rotate your initial solution. In general, I think there is a need of more details.  
  2. I’m wondering whether it could have been better to split your big sample into two parts: a first sample in which you could explore the structure of your measure, and a validation sample in which you could confirm the structure by applying confirmatory factor analysis. Did you consider this possibility?
  3. I believe that the correlation between your new measure and work-family conflict, as well as family-to-work conflict, cannot be considered small. It is .31 with WFC and .28 with FWC, and both are stat. significant. It is also not clear why the above evidence should be taken as proof of divergent validity.  

Reviewer 2 Report

Comments and Suggestions for Authors

The study presents the development of a novel scale to measure ambivalence both within healthcare professionals and general public. The manuscript is well written and the sample is adequate for the conducted analysis. The results are potentially useful for the field. However, there are areas that should be expanded or clarified to warrant publication. Main issues are listed below:

-If the information has been collected, please describe the sample demographics in more detail, including mean age and the distribution with regards to healthcare profession.

-Please include the original german version of the scale as a table or a supplement. 

-Please discuss the differences of the obtained sample from the general population and the applicability of the results for both groups.

-Was the data screened for quality of answering (random, monotonous etc.)? If not, discuss as a limitation

-Please provide more details on EFA. What extraction method was used? No rotation was performed? Also, additional test of factor amount would increase rigot (Velicer MAP, VSS, etc.)

-Discuss the basis for usage of 4-step Likert scale as ordinal, compared to categorical. 

-Including rotated loadings alond with 2+ factor solution loadings as a table would lend credibility to the single factor solution as being optimal. This could be included as supplemental material. 

-Include item means, standard deviations and ranges in Table 1.

-Divergent validity from the Schneider et al. 2021's existing scale would be highly interesting to readers. As it has not been included in the data collection, I understand this analysis can not be included. However, it should receive attention in the discussion. Why should use the scale presented here instead? What is the evidence for it?

-Invariance analysis is missing. Does the scale function similarly between age groups and gender? Additionally, some form of medical profession variable would be interesting here.

-Together with invariance analysis, average scores in terms of demographics should be reported.

-In general, discussion section is very limited and should be expanded. Future work needs should be spelled out explicitly and in more detail.

If these issues are addressed, I am glad to re-review the manuscript. 

Round 2

Reviewer 1 Report

Comments and Suggestions for Authors

Thank you for improving the manuscript according to my suggestions. I have some remaining issues, which I report below. 

  1. p. 4 line 175. The acronyms EFA etc should be spell out (e.g., EFA-Exploratory factor analysis) given in full.
  2. p. 7. Can you explain with a sentence or two why it is important to test measurement invariance on gender and age?
  3. p. 7, line 295 Can you give a reference for the invariance threshold used?
  4. p. 7, from line 296. You provide here the invariance results. I think the results should be reported in the Results section.
  5. Table 1. Students were also involved. But students is not a professional group and in the Data collection section (p. 3) you reported that being employed was an inclusion criteria. Can you clarify?
  6. p. 9. Confirmatory factor analysis section, line 373. There is a missing letter at the beginning of the paragraph.
  7. p. 9, from line 375. You have already mentioned fit indices above (see Statistical analysis section). There is no need to mention them again.

Author Response

Please enter "Please see the attachment.

Reviewer 2 Report

Comments and Suggestions for Authors

My concerns have been addressed and I think the manuscript has improved substantially. Only one minor point remains: I find the term "legacy scale" a bit strong, suggesting that the older instruments' obsolescent . I suggest replacing these instances with something more neutral, such as existing, previous etc.

Author Response

(The authors gave the same response as above.)
